# Production of a Foot-and-Mouth Disease Vaccine Antigen Using Suspension-Adapted BHK-21 Cells in a Bioreactor

**DOI:** 10.3390/vaccines9050505

**Published:** 2021-05-13

**Authors:** Soonyong Park, Ji Yul Kim, Kyoung-Hwa Ryu, Ah-Young Kim, Jaemun Kim, Young-Joon Ko, Eun Gyo Lee

**Affiliations:** 1Bioprocess Engineering Center, KRIBB, 30 Yeongudanjiro Ochang-eup, Chungju-si 28116, Korea; sypark@kribb.re.kr (S.P.); jykim@kribb.re.kr (J.Y.K.); khryu@kribb.re.kr (K.-H.R.); redoorkim@kribb.re.kr (J.K.); 2Department of Bioprocess Engineering, Korea University of Science and Technology (UST), Daejeon 34113, Korea; 3Center for FMD Vaccine Research, Animal and Plant Quarantine Agency, Gimcheon 39660, Korea; mochsha@korea.kr (A.-Y.K.); koyoungjoon@korea.kr (Y.-J.K.)

**Keywords:** adherent, baby hamster kidney cell, foot-and-mouth disease, virus productivity, suspension culture, vaccine

## Abstract

The baby hamster kidney-21 (BHK-21) cell line is a continuous cell line used to propagate foot-and-mouth disease (FMD) virus for vaccine manufacturing. BHK-21 cells are anchorage-dependent, although suspension cultures would enable rapid growth in bioreactors, large-scale virus propagation, and cost-effective vaccine production with serum-free medium. Here, we report the successful adaptation of adherent BHK-21 cells to growth in suspension to a viable cell density of 7.65 × 10^6^ cells/mL on day 3 in serum-free culture medium. The suspension-adapted BHK-21 cells showed lower adhesion to five types of extracellular matrix proteins than adherent BHK-21 cells, which contributed to the suspension culture. In addition, a chemically defined medium (selected by screening various prototype media) led to increased FMD virus production yields in the batch culture, even at a cell density of only 3.5 × 10^6^ cells/mL. The suspension BHK-21 cell culture could be expanded to a 200 L bioreactor from a 20 mL flask, which resulted in a comparable FMD virus titer. This platform technology improved virus productivity, indicating its potential for enhancing FMD vaccine production.

## 1. Introduction

Foot-and-mouth disease (FMD) is a highly contagious acute disease with significant economic impacts; it affects almost all types of cloven-hoofed domestic and wild animals globally. The disease presents with fever, lameness, and vesicular lesions on the foot and tongue. The causative agent, foot-and-mouth disease virus (FMDV), is a positive-sense, single-stranded RNA virus belonging to the Picornaviridae family. This virus has seven antigenically variable serotypes (O, A, C, Asia1, SAT1, SAT2, and SAT3) as well as various subtypes [1].

Several attempts have been made to develop FMD vaccines to prevent infection. The first inactivated FMD vaccine was developed by Waldmann et al. in 1937, using virus from vesicular fluid obtained from the tongues of deliberately infected cattle [2]. In the 1950s, Frenkel devised a large-scale in vitro culture technology for virus production in slices of surviving bovine tongue epithelium [3]. However, Frenkel’s method is difficult to scale up due to a limited supply of tongue epithelium. To overcome these limitations, further research was initiated to identify a cell line susceptible to FMDV infection. Mowat and Chapman demonstrated that BHK-21, clone 13, previously developed by Macpherson and Stocker in 1962 was capable of supporting FMDV growth [4,5,6]. For industrial-scale vaccine production, Capstick et al. (1962) adapted BHK-21 for growth in suspension, and Telling and Elsworth (1965) used the adapted BHK-21 cells to produce an inactivated FMD vaccine in a large-scale bioreactor in serum-containing medium [7,8].

However, culturing cells in medium with serum has some disadvantages [9,10]. First, serum may contain potential factors such as bacteria, fungi, viral contaminants, mycoplasma, prion proteins, endotoxins, and unknown components that contaminate a cell culture and interfere with downstream processing and product purification. Serum also has lot-to-lot qualitative, quantitative, geographical, and seasonal variations, which lead to inconsistent cell growth and productivity. In addition, serum is not economical for large-scale bioreactor culture due to the high cost of serum.

Large-scale virus production with adherent mammalian cells is feasible using a roller bottle [11], but suspension-culture systems grown in shake flasks and stirred in a bioreactor are easier to scale up and control, and have high volumetric productivity due to higher maximum viable cell densities (MVCDs) than are possible when compared to adherent cells [12]. The simplest strategy for suspension adaptation is a direct method that involves the direct change from an adherent-culture condition with serum-containing medium to an agitation-culture condition with serum-free medium. However, interruption of cell-surface protein signaling, such as the loss of integrin signaling after removing cell-adhesive surface proteins and disrupting extracellular matrix (ECM)–cell contacts, leads to a specific form of apoptosis called anoikis [13].

Cell death can occur after removing adhesive surface proteins or serum withdrawal. Serum contains diverse components with biological functions, such as growth factors, cytokines, supplemental metabolites, transport proteins, and hormones [14], as well as physiological factors that function in pH buffering, toxin and protease inactivation, and protection from shear stress [15]. Moreover, serum has an anti-apoptotic effect that is independent of nutrient deprivation [16]. To address the problem of cell aggregation and apoptosis caused by direct adaptation, we incrementally reduced the serum concentration while maintaining high viability (>80%) during adaptation. This gradual reduction of serum over time increases the probability of successful suspension adaptation, since even a small amount of serum that remains during adaptation may contribute to adapting the new environment. During the adaptation process, it is important to monitor the cell-growth rate since an increase is the first sign of successful adaptation. To achieve successful adaptation with increased cell growth, it is common to add supplements that support cell growth, such as commercial serum replacements or defined feeding media.

Here, we developed suspension-adapted BHK-21 (BHK-S) cells capable of growth to a high cell density in serum-free medium and evaluated the adhesive properties of the suspension-adapted cells. We also developed serum-free medium for high-yield FMDV production. Suspension cell cultures were scaled up from flasks to 200 L bioreactors. FMDV can be divided into several particles (146S, 75S, and 12S) and 146S is intact virions [17]. Since 146S antigens are the most efficient immunogens in the FMD vaccine, 146S antigens were also quantified in each production scale [18]. This cell-culture-based FMDV-production technology can be employed to develop an effective viral vaccine within a short period against newly spreading FMDV strains.

## 2. Materials and Methods

### 2.1. Cell Lines, Cell Culture Media, and FMDV

An adherent BHK-21 cell line (C-13, ATCC CCL-10) was grown in T-75 flasks at 37 °C and 5% CO_2_ in Dulbecco’s modified Eagle’s medium (DMEM; Gibco, Grand Island, NY, USA) supplemented with 10% fetal bovine serum (FBS; Gibco, USA). The culture was diluted and passaged every 3 or 4 days. The VCD and cell viability were measured by trypan blue exclusion using a Vi-CELL XR analyzer (Beckman Coulter, Fullerton, CA, USA). FMDV O/Jincheon/SKR/2014 (Mya-98 lineage strain) was used in this study for virus titration with adherent and suspension adapted BHK-21 cells [19].

### 2.2. Suspension Adaptation of Adherent BHK-21 Cells and Maintenance of a BHK-21 Suspension (BHK-S) Culture

Adherent BHK-21 cells were directly adapted to ProVero-1 medium (Lonza, Basel, Switzerland) containing 10% *v/v* FBS, 4 mM l-glutamine, 5% *v/v* homemade feed medium-1 (FM-1; 25 g/L, pH 7.4), 1 g/L PF-68, and 0.2 g/L dextran sulfate (Sigma-Aldrich, St. Louis, MO, USA) in Erlenmeyer flasks. The cells were incubated at 37 °C and 5% CO_2_, with agitation at 110 rotations/min (rpm), and FBS was progressively eliminated during the weaning procedure. Once the cells were adapted to growth in suspension, the resulting BHK-S cells were serially passaged every 3 or 4 days until they showed stable growth in the serum-free medium. The suspension-adapted cells were cryopreserved at different passages for analysis, and fully adapted BHK-S cells were deposited in the Research Cell Bank under serum-free conditions (KCTC 12945BP, KCTC18706P; Korean Collection for Type Cultures, Korea Research Institute of Bioscience and Biotechnology).

### 2.3. Cell-Adhesion Assay

Cell adhesion assay was performed under sterile conditions using the CytoSelect 48-Well Cell Adhesion Assay, ECM Array (Cell Biolabs, San Diego, CA, USA). Briefly, the plates coated with various ECM components (fibronectin, collagen I, collagen IV, fibrinogen, and laminin or bovine serum albumin as a negative control) were warmed up at room temperature for 10 min. 1 × 10^6^ cells/mL cells were then prepared in serum-free medium, 150 μL of cell suspension was added to appropriate wells, and the plates were incubated for 60 min at 37 °C in a humidified atmosphere in the presence of 5% CO_2_. The medium was removed, and each well was washed 4 times with 250 μL phosphate-buffered saline (PBS). The PBS was removed and 200 μL of the cell-staining solution (provided in the kit) was added to each well and the plates were incubated for 10 min at room temperature. After incubation, the cell-staining solution was removed, and each well was washed 4 times with 500 μL deionized water. The final wash was removed, and the wells were dried. Then, 200 μL extraction solution (provided in the kit) was added to each well, and the plates were incubated for 10 min on an orbital shaker at room temperature. Finally, 150 μL of each sample was transferred to each well in a 96-well microtiter plate (Thermo Fisher Scientific, Waltham, MA, USA), and the optical density at 560 nm (OD_560nm_) was measured using a Synergy HT (BioTek Instruments, Winooski, VT, USA). Each experiment was carried out in triplicate.

### 2.4. Development of Serum-Free Medium for BHK-S Cells

A prototype media-formulation library was prepared using a design-of-experiments-based approach, which is supported by Thermo Fisher Scientific (Waltham, MA, USA). Twenty milliliters of each base medium was added into a 125 mL Erlenmeyer flask, followed by 0.3 × 10^6^ BHK-S cells/mL, after which the cells were incubated at 37 °C and 5% CO_2_ with shaking at 110 rpm. Sixteen prototype media (eight chemically defined and eight protein-free media) and ProVero-1 medium were screened in triplicate after adaptation for three passages. BHK-S cells in each medium were analyzed to assess their growth kinetics every 24 h for 6 days, which enabled us to determine the maximum viable cell concentration reached for each experimental condition tested. After developing serum-free media, BHK-S cells were adapted to the new environment until cell growth and viability were stable.

### 2.5. Propagation and Titration of FMDV

FMDV vaccine strain O/Jincheon/SKR/2014 (Mya-98 lineage strain) were obtained from Jincheon County (Korea) in 2014. Virus propagation was conducted in a biosafety-level 3 containment facility at the Animal and Plant Quarantine Agency in South Korea. On day 3 after passaging adherent BHK-21 cells and BHK-S cells, the medium was changed and the cells were infected with FMDV O/Jincheon/SKR/2014 (Mya-98 lineage strain) at a multiplicity of infection (MOI) of 0.01 or 0.001. Supernatants from infected cultures were harvested at 16, 20, or 24 h post-infection (hpi). FMDV titers were determined in the adherent BHK-21 cells via an endpoint titration using the Spearman–Kärber calculation and were presented as the tissue culture infective dose affecting 50% of the cultures (TCID_50_) per mL [20,21]. Each experiment was carried out in triplicate.

### 2.6. 146S Antigen Quantification

The clarified virus culture supernatant of 50 mL was inactivated with 3 mM binary-ethylenimine (Sigma-Aldrich) treatment at 26 °C for 24 h and concentrated by polyethylene glycol (PEG, MW 6000, Sigma-Aldrich) precipitation. The precipitate was resuspended in tris-KCl (TK) buffer (pH 7.6) and then layered onto 15–45% sucrose density gradients (SDGs) and ultracentrifuged at 100,000× *g* for 4 h at 4 °C using a SW41Ti rotor(Beckman Coulter, Brea, CA, USA). The ultra-centrifuged SDG was fractionated using a continuous density gradient fractionator (Teledyne ISCO, Lincoln, NE, USA) [22], and the absorbance of each fraction at 254 nm was consecutively recorded with the spectrophotometer component of the instrument. The area under the peak for specific fractions was measured to calculate the quantity of 146S antigens (µg/mL) according to a previous study [22]. Each experiment was carried out in triplicate.

### 2.7. Optimizing BHK-S Batch Culture

BHK-S cells were expanded in various culture modes, including batch, fed-batch, and enhanced-batch cultures. In the fed-batch culture, a 5% *v/v* of FM-1 (25 g/L, pH 7.4) was added to chemically defined medium (CDM)-2 on day 0 and day 2. Enhanced batch cultures were tested by adding FM-1 (25 g/L, pH 7.4) to the initial medium at a ratio of 5, 10, 15, or 20% (*v/v*). Each experiment was carried out in triplicate.

### 2.8. Scaled Up FMDV Production in 50 L and 200 L Bioreactors

A scale-up study was performed using stirred tank bioreactors. Briefly, a 50 L and a 200 L bioreactor were filled with 50 L and 200 L CDM-2, respectively, inoculating 0.5 × 10^6^ BHK-S cells/mL. The temperature, dissolved oxygen level, and pH in the bioreactor were controlled at 37 °C, 40% air saturation, and 7.2 ± 0.2, respectively. The agitation speeds of the impellers in the 50 and 200 L bioreactors were 80 and 50 rpm, respectively. After 3 days of culture, the BHK-S cells were allowed to settle down without agitation for 12 h, and 80% of the old medium was replaced with fresh CDM-2. After the medium exchange, the cells were infected with FMDV at an MOI of 0.01, and the infected cultures were harvested at 16 hpi for FMDV quantification.

## 3. Results

### 3.1. Suspension Adaptation and BHK-S Cell Growth

Before suspension adaptation, we screened various commercial serum-free media in Erlenmeyer flasks to select an optimal serum-free medium for suspension adaptation of BHK-21 cells, with active cell growth and fewer aggregates. ProVero-1 medium was found to be optimal for BHK-21 growth. The commercial serum-free media tested contained abundant components as serum replacements, several of which are biologically active, such as growth factors, cytokines, supplemental metabolites, transport proteins, and hormone and survival factors that elicit anti-apoptotic and protective effects independently of nutrient deprivation. However, under stressed conditions (removal of serum or adhesive surface proteins, or exposure to shear stress), cell cycle arrest occurred and cell growth was delayed [23,24]. Therefore, homemade FM-1 feed, which contained nutrients such as lipids, amino acids, and vitamins, was added to support cell growth [25].

Adherent BHK-21 (C13) cells were directly adapted to ProVero-1 medium in Erlenmeyer flasks. The FBS was eliminated by stepwise weaning from 10 to 0%. During suspension adaptation, the BHK-21 cells easily formed aggregates and attached to the culture vessel due to hydrodynamic effects [26]. These aggregates resulted from DNA released from dead cells due to a failure to adapt to suddenly changed culture conditions and can significantly influence nutrient uptake, the cell-proliferation rate, shear stress, and cell death, which are ultimately detrimental to the cell culture process [27]. Surfactant PF-68 is widely used in mammalian cell cultures to protect cells from shear stress, and dextran sulfate is also used as an anti-aggregation agent [28,29]. Therefore, we used both reagents in combination to efficiently prevent cell aggregation, and we removed formed cell aggregates by filtering.

The complete adaptation to serum-free suspended cultivation took 150 days, and the resulting cell line was designed as the BHK-S cell line. BHK-S cell morphology was compared to adherent BHK-21 cells grown in DMEM with 10% FBS (Figure 1A). The BHK-S cells showed a typical round shape without aggregate formation, and the morphology was retained during several passages (Figure 1B). After suspension adaptation, BHK-S cells had stable cell growth and viability over 30 days (Figure 1C), during which time the VCD reached 7.65 (±0.71) × 10^6^ cells/mL on day 3. Compared to BHK-21 cells, BHK-S cells showed a comparable doubling time and growth rate, and a dramatically higher volumetric VCD (Table 1). To compare FMDV productivity between BHK-21 and BHK-S cells, they were infected with FMDV in T-75 flasks and 125 mL Erlenmeyer flasks, respectively. The BHK-S cells generated a 5.6-fold higher virus titer than the BHK-21 cells, due to volumetric productivity (Figure 1D). Finally, BHK-S cells having robust cell growth and lower aggregate formation were banked as a host cell line for viral infection.

### 3.2. ECM Properties of BHK-S Cells

The ECM is one factor that determines the adhesive properties of cells cultured in suspension. To evaluate whether the BHK-S cells had adhesive properties, we performed cell-adhesion assays for five ECM proteins (fibronectin, collagen I, collagen IV, laminin I, and fibrinogen) with adherent BHK-21 cells, intermediately adapted BHK-S cells (grown in medium containing 5% FBS), and BHK-S cells (Figure 2). The intermediately adapted BHK-S cells (in 5% FBS) and BHK-S cells exhibited dramatically reduced adhesion to all five ECM proteins, when compared to BHK-21 cells. The intermediately adapted BHK-S cells showed slightly higher adhesion to all five ECM proteins, except for collagen Ⅰ, when compared to BHK-S. Consequently, BHK-S cells showed significantly lower adhesion characteristics in serum-free medium, which likely contributed to suspension adaptation. Additionally, BHK-S cell aggregates gradually disappeared and the suspension ratio increased from 67.5 (±0.4)% to 93.5 (±3)%.

### 3.3. FMDV Susceptibility and Productivity of BHK-S Cells

To evaluate whether virus sensitivity remained after suspension adaptation, BHK-S cells were infected at a density of 6 × 10^6^ cells/mL with FMDV (MOI = 0.01 or 0.001) immediately after fresh medium replacement. FMDV was propagated for 16, 20, and 24 hpi, and the cytopathic effect increased to >90%. At different MOIs, the FMDV titers were similar, but decreased at increasing hpi. Therefore BHK-S cells retained efficient sensitivity to FMDV infection, even at an MOI of 0.001, although the production yield of FMDV diminished over time (Figure 3A).

To compare the volumetric productivity of FMDV, we evaluated the correlation between the cell density and FMDV productivity. BHK-S cells were infected at different cell concentrations (approximately 1 × 10^5^, 1 × 10^6^, and 1 × 10^7^ cells/mL). As the cell concentration increased, FMDV productivity also increased (Figure 3B).

### 3.4. Developing Serum-Free Medium for BHK-S Cells

To develop serum-free medium for enhanced FMDV production, we screened 16 types of serum-free and animal-component-free prototype media for BHK-S cells, which were obtained from Thermo Fisher Scientific, Inc. The prototype media consisted of two groups, including CDM and hydrolysate-derived protein-free media (PFM). The optimal serum-free medium was selected, based on both the cell-growth rate and the virus titer. The VCDs of BHK-S cells in ProVero-1 medium (control medium) and 16 serum-free prototype media were determined at days 3, 4, and 5 (Figure 4A). Among the CDMs, CDM-5 showed the highest maximum viable cell concentration (MVCC) with BHK-S cells on day 4, which was approximately 7.1 × 10^6^ cells/mL. BHK-S cells grown in PFM-2 reached a density of 11.22 × 10^6^ cells/mL on day 4 and 19.06 × 10^6^ cells/mL on day 5, resulting in the highest MVCC among all prototype media tested. The FMDV titers in these prototype media were also evaluated (Figure 4B). Effective FMDV productivity was achieved with BHK_S cells grown in CDM-2, CDM-3, and CDM-4, and specifically, CDM-2 had the highest FMDV productivity of 17 × 10^6^ TCID_50_/mL. Although most PFMs facilitated higher cell growth than CDMs, the virus titers were higher with the CDM prototypes. Although the MVCC of cells grown in PFM-2 was 4-fold higher than that of cells grown in CDM-2, cells grown in CDM-2 produced 6.6-fold more virus than cells grown in PFM-2. Consequently, CDM-2 was selected as the optimum serum-free medium for high-yield FMDV production.

### 3.5. Optimizing BHK-S Batch Culture and Scaled-Up FMDV Production in a Bioreactor

Culture modes were developed to enhance the cell growth rate. To simplify the cultivation process and minimize contamination, the initial CDM-2 medium was supplemented with various feeding concentrations for enhanced batch culture. As the concentration of FM-1 increased from 5 to 20%, the MVCC were increased in a dose dependent manner, whereas 20% FM-1 dramatically suppressed cell growth (Figure 5). BHK-S cells grown in CDM-2 containing 15% FM-1 reached a density of approximately 11.5 (±1.34) × 10^6^ cells/mL on day 4, which represented a higher MVCC than the fed-batch culture, and high viability was also sustained on day 5.

Scaled-up cultures were performed in both 50 and 200 L bioreactors to confirm whether FMDV production was applicable in a large-scale bioreactor (Table 2). In the 50 L bioreactor, the cell density was similar to that of the flask on day 3, but in the 200 L bioreactor, the cell density was lower than that in the smaller bioreactor. Nevertheless, the virus titer was 11 × 10^6^ TCID_50_/mL in the 200 L bioreactor, higher than that in the 50 L bioreactor, and the 146S antigen concentrations were 2.56 and 2.1 μg/mL in the 50 and 200 L bioreactors, respectively.

## 4. Discussion

Suspension cell culture in serum-free medium is a major platform for the commercial production of therapeutics in mammalian cell cultures because this approach reduces the production cost, eliminates pathogens from animal-component sources, and shows consistent cell-based performance. However, most mammalian cells, such as Chinese hamster ovary (CHO) and BHK-21 cells originally required anchorage and serum for vigorous cell proliferation. In this study, we developed a suspension BHK-21 cell line and animal-component-free CDM for FMDV vaccine production.

In addition to morphological changes, we observed changes in the cell-adhesion properties of BHK-S cells. Some methods are available for analyzing differently expressed genes (DEGs) related to cell adhesion after suspension adaptation, such as microarray and qRT-PCR (quantitative real time-polymerase chain reaction) analysis [30,31,32]. Instead of DEG analysis, we performed cell-adhesion assays with five types of ECM protein to evaluate changes in the cell-adhesion properties of adherent and suspension-adapted BHK-21 cells [33]. Suspension-adapted BHK-21 cells showed significantly lower adhesion to all five ECM proteins than wild-type adherent BHK-21 cells (Figure 2). We suggest that during suspension adaptation, for survival and proliferation in serum-free conditions, the adhesive properties of the cells decreased through integrin rearrangements on the cell membrane and the actin cytoskeleton [34,35]. The microenvironment is a fundamental regulator of cellular behavior, and the ECM is a key factor in the microenvironment that is necessary for cell adhesion. Cellular interactions with the ECM occur primarily through the integrin family of cell-surface receptors, which are activated by cytoskeletal components [36,37]. Interactions between the ECM and integrins activates inside-out and outside-in signaling pathways that control cell behaviors such as cell-cycle arrest and proliferation [38,39]. However when anchorage-dependent cells detach from the ECM, they can undergo anoikis [40,41]. Occasionally, detached cells gain resistance to anoikis and adapt to their new environment, after which they have probably become anchorage-independent and finally survive in the absence of signaling from the ECM. This phenomenon is one of the hallmarks of cancer cells [42,43,44]. Indeed, malignantly transformed cells secrete their own ECM ligands, enabling them to escape proliferative growth suppression and to survive in an anchorage-independent manner [45]. Furthermore, cancer-related genes may be related with the suspension adaptation of CHO-K1 and HEK293 cells [46,47].

In addition to growth, morphological changes of BHK-S cells, and their ECM characteristics, the productivity of cells is another important parameter. FMDV infects host cells using integrins as receptors. The interaction between FMDV and an integrin molecule is mediated by an Arg–Gly–Asp triplet located at the G–H loop of the capsid protein, VP1. FMDV isolates interacting with integrins gain cell entry following clathrin-mediated endocytosis, which induces internalization [48]. Therefore, since changes in the adhesive properties of BHK-S cells can influence FMDV infectivity, we evaluated the FMDV sensitivity of BHK-S cells. Even at a low MOI of 0.001, BHK-S cells had a similar titer versus BHK-S cells infected at an MOI of 0.001–0.01 (Figure 3A); thus, suspension cultures of BHK-S cells displayed advantages in terms of high cell densities and volumetric productivities, facilitating large-scale virus production and cost-effective vaccine production.

Suspension cultures in bioreactors, which enable cultivation to higher cell densities, can increase the viral production yield and offer cost-effective and easy scale-up with serum-free medium or CDM. However, with various virus production systems, the specific virus yield per cell can be lower than expected in proportion to the cell density, an effect referred to as the “cell-density effect” [49,50]. This phenomenon is thought to be caused by a limitation of key nutrients, by-product inhibition, or cell-cycle arrest at high cell densities, although the exact mechanism(s) involved remain unknown [51,52]. Some strategies have been developed to overcome the cell-density effect, such as perfusion culture and medium replacement before virus infection [53,54,55,56]; thus, we performed batch mode cultivation with fresh medium replacement with the aim of developing a simple process. We tested 16 prototype media (eight CDMs and eight PFMs) and screened CDM-2, which showed the highest virus titer even at a low cell density (Figure 4). Most PFMs enabled growth to higher cell densities, but showed lower virus productivity than the CDMs, suggesting that PFM may contain components that inhibit virus infection or propagation. Our findings indicate that innovative media development can facilitate high cell-density cultivation and high viral production [57].

Scaled-up mass culture using a bioreactor can monitor and control detailed parameters for cell growth, such as pH, dissolved oxygen, temperature, and agitation rate. However, in the development of a scale-up process for a bioreactor from a flask, several variables must be considered. In this study, scaled-up production for FMDV using enhanced batch culture was comparable from a flask to a 300 L bioreactor. Because of process parameters, such as agitation rate, cell growth in the 50 L bioreactor was higher than that in the flask or 300 L bioreactor, but the quantitative values of 146S antigen were similar.

## 5. Conclusions

In this study, we successfully developed BHK-21 suspension cells and CDM for producing FMDV vaccines in a large-scale bioreactor. The BHK-S cells showed lower ECM-protein adhesion than adherent BHK-21 cells, and cell aggregation was dramatically reduced. To increase FMD vaccine productivity, a customized medium was developed, which made it possible to determine the medium components required for rapid cell expansion of the medium platform and to secure price competitiveness. Scaling up from a 50 L bioreactor to a 200 L bioreactor showed comparable cell growth and FMDV titers, when compared to flask cultures. Finally, we developed an efficient cell-based FMDV vaccine-production platform using BHK-21 suspension cells grown in CDM, which can potentially be applied to develop an effective viral vaccine in the case of a new FMD outbreak.

## Figures and Tables

**Figure 1 vaccines-09-00505-f001:**
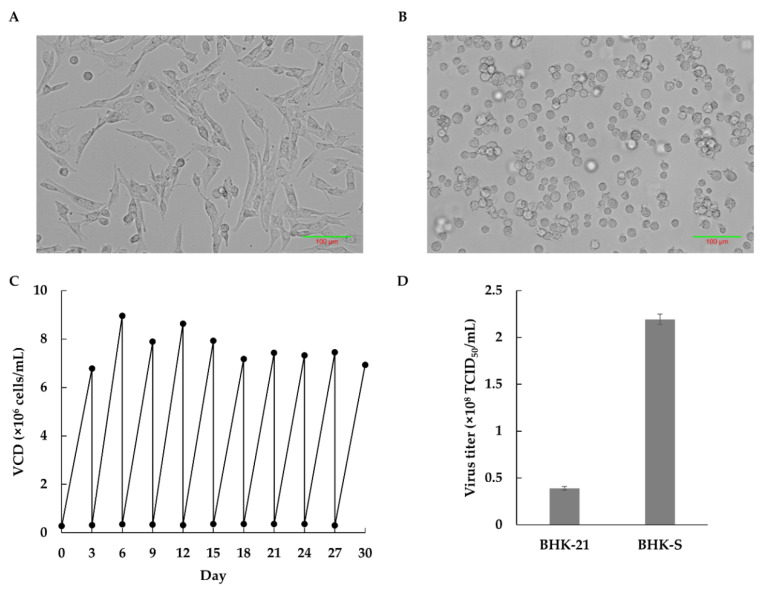
Morphologies of adherent BHK-21 (C13) cells in a T-flask (**A**) or BHK-S cells as suspension cultures in a 125 mL Erlenmeyer flask (**B**). The stability of the BHK-S cells was monitored for 3 weeks (**C**). The cell viability exceeded 95% at each passage. Production of FMDV by BHK-21 and BHK-S cells was compared using the same culture volume (**D**). Each experiment was carried out in triplicate. Data expressed as mean ± standard deviation.

**Figure 2 vaccines-09-00505-f002:**
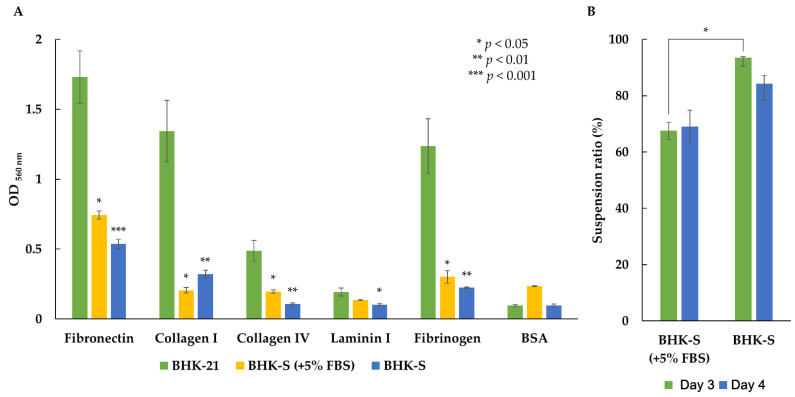
Suspension BHK-S cells and intermediately adapted BHK-S cells (+5% FBS) showed reduced cell attachment to the ECM proteins fibronectin, collagen type I, collagen type IV, laminin I, and fibrinogen (**A**). Colorimetry was performed to quantify the OD _560 nm_ values after cell staining and extraction. The BHK-S cells showed less aggregation than the intermediately adapted BHK-S cells (+5% FBS) (**B**). Each experiment was carried out in triplicate. Data expressed as mean ± standard deviation. Statistical analyses were performed using Student’s *t*-test. *p* < 0.001 was considered statistically significant.

**Figure 3 vaccines-09-00505-f003:**
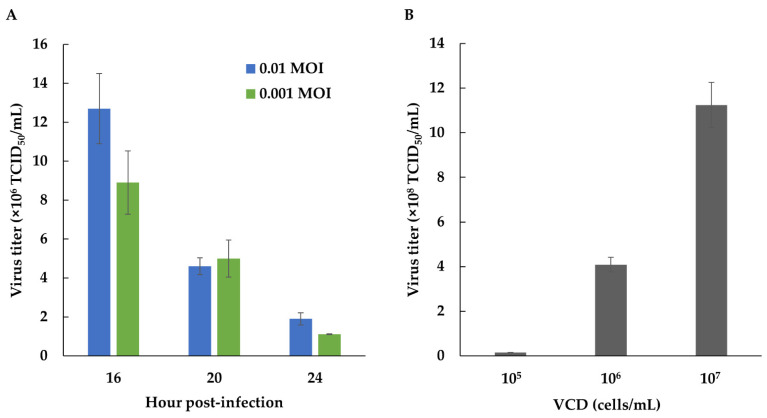
The sensitivity of BHK-S cells to FMDV infection was examined using FMDV MOIs of 0.01 and 0.001 (**A**). Effect of a high cell density on FMDV production (**B**). BHK-S cells were infected with an FMDV MOI of 0.01, at viable cell densities of 10^5^, 10^6^, and 10^7^ cells/mL. Each experiment was carried out in triplicate. Data expressed as mean ± standard deviation.

**Figure 4 vaccines-09-00505-f004:**
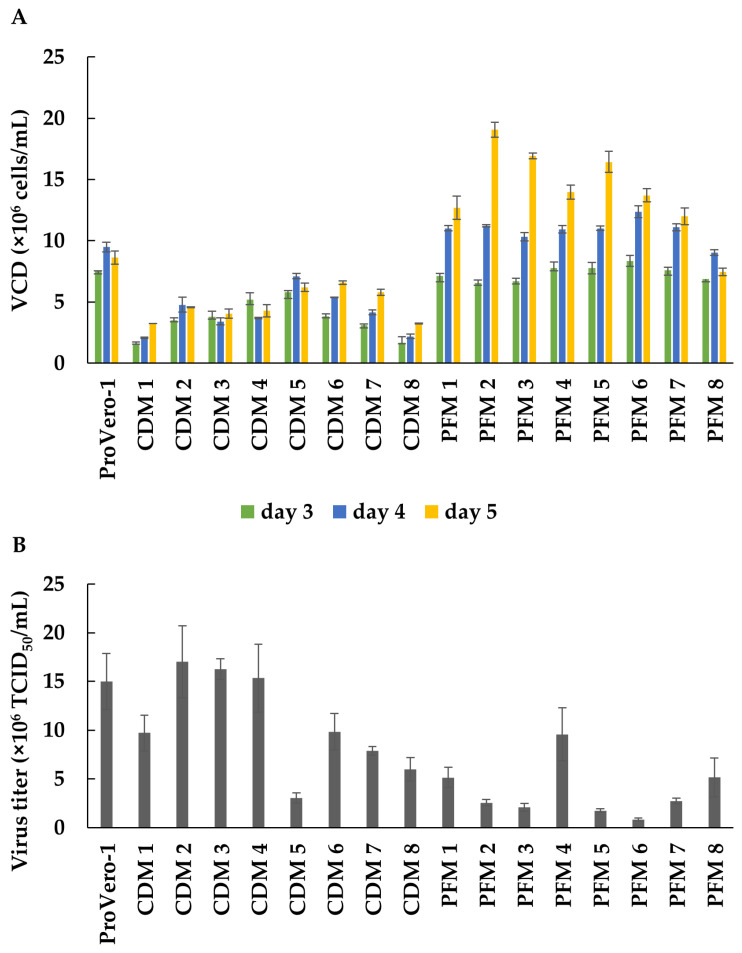
Serum-free medium was developed for increasing FMDV productivity. The growth profile of BHK-S cells was monitored using eight CDM prototypes and eight PFM prototypes (**A**). FMDV titers were measured on the third day of BHK-S cell culture using the indicated prototype media (**B**). Each experiment was carried out in triplicate. Data expressed as mean ± standard deviation.

**Figure 5 vaccines-09-00505-f005:**
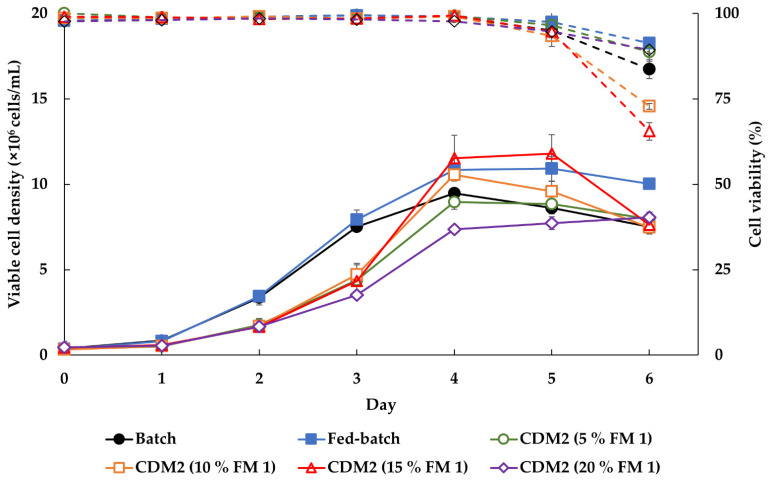
Various culture modes were tested to increase cell density of BHK-S cells and to simplify the culture process. BHK-S cell growth was compared between batch culture (●), fed-batch culture (■), and enhanced batches grown in CDM-2 supplemented with 5 (**○**), 10 (**□**), 15 (△), and 20 (◇)% home-made FM- 1 (25 g/L, pH 7.4). On days 0 and 2, the media used for the fed-batch cultures were supplemented with 5% *v/v* FM-1. Solid line is viable cell density and dash line is cell viability. Each experiment was carried out in triplicate. Data expressed as mean ± standard deviation.

**Table 1 vaccines-09-00505-t001:** Characteristic comparison between adherent BHK-21 (C13) and BHK-S.

	VCD at Day 3 (10^6^ cells/cm^2^ or cells/mL)	Doubling Time (h)	Specific Growth Rate (h^−1^)
BHK-21 (C13)	0.45 (±0.02)	15.53 (±0.69)	0.045 (±0.002)
BHK-S	7.65 (±0.71)	15.97 (±0.57)	0.043(±0.002)

**Table 2 vaccines-09-00505-t002:** Scaled-up production of FMDV.

Scale-Up	Cell Density at Infection (cells/mL)	Virus Titer (TCID_50_/mL)	146S Antigen (μg/mL)
250 mL Flask	7.98 (±0.14)	25.00 (±2.35)	2.30 (±0.28)
50 L bioreactor	10.83	9.70	2.56
200 L bioreactor	6.58	11.00	2.10

## Data Availability

Data available upon request.

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
