# Peer review of "Production of a Foot-and-Mouth Disease Vaccine Antigen Using Suspension-Adapted BHK-21 Cells in a Bioreactor"

_vaccines, 2021, doi:10.3390/vaccines9050505_

Round 1
Reviewer 1 Report
The manuscript by Park et al describes a systematic protocol for the efficient adaptation of BHK-21 to grow as a suspension and under serum-free conditions media. The manuscript reinforces the value of using this cell line for large-scale preparation of FMDV vaccines, specifically for their use in bioreactors. With the exception of a few typos, the manuscript is well-written. However, some clarifications are required in the methodology and the incorporation of a critical control in Fig. 3. Please see my comments below:
Line 62: “The simplest strategy of suspension adaptation is [a ]direct method that involves the change from…
Line 75-76: The statement implies that serum-free conditions facilitate suspension adaptation. The sentence is confusing and the authors need to rephrase since this paragraph focuses on describing the importance of a stepwise reduction in serum for successful adaption of BHK-21 to serum-free media.
Line 151-153: the authors need to specify how much of the clarified supernatant was used to measure 146 S antigen (especially from the 50L and 250L bioreactor scales). When describing PEG precipitation the authors need to include the molecular weight of the PEG.
Figure 3: the authors set up a few experiments with different moi to examine the sensitivity of BHK-S to FMDV. However, the authors failed to add the conventional adhesive culture of BHK-21 cells at the same cellular density as a control. The fact that yield of FMDV is reduced over time may suggest that suspension and serum-free adaptation reduced the expression of some molecules (i.e., integrins) required for virus entry.
In all graphs the authors need to indicate how many experiments were conducted and whether the error bars one standard deviation of uncertainty or one standard error.
Author Response
Reviewer : The manuscript by Park et al describes a systematic protocol for the efficient adaptation of BHK-21 to grow as a suspension and under serum-free conditions media. The manuscript reinforces the value of using this cell line for large-scale preparation of FMDV vaccines, specifically for their use in bioreactors. With the exception of a few typos, the manuscript is well-written. However, some clarifications are required in the methodology and the incorporation of a critical control in Fig. 3. Please see my comments below:
Reviewer comment 1: Line 62: “The simplest strategy of suspension adaptation is [a ]direct method that involves the change from…:
Author response 1: This sentence has been corrected according to the reviewer’s comment, as follows: “The simplest strategy of suspension adaptation is a direct method that involves the change from…”
Reviewer comment 2: Line 75-76: The statement implies that serum-free conditions facilitate suspension adaptation. The sentence is confusing, and the authors need to rephrase since this paragraph focuses on describing the importance of a stepwise reduction in serum for successful adaption of BHK-21 to serum-free media.
Author response 2: The statement has been clarified to focus on the importance of a stepwise reduction in serum.
Line 75 on page 2 : This gradual reduction of serum over time increases the probability of successful suspension adaptation, since even small amount of serum that remained during adaptation, may contribute to adapt the new environment.
Reviewer comment 3: the authors need to specify how much of the clarified supernatant was used to measure 146S antigen (especially from the 50L and 250L bioreactor scales). When describing PEG precipitation the authors need to include the molecular weight of the PEG.
Author response 3: We used 50 mL of the clarified supernatant to measure 146S antigens for all culture scales, and the molecular weight of PEG in antigen precipitation was 6,000 from Sigma. This information has been added to page 4, line 165 and 167. This information was added to Line 165 and Line 167 on page 4.
Line 165~167 : The clarified virus culture supernatant of 50 mL was inactivated with 3 mM binary-ethylenimine (Sigma-Aldrich, St. Louis, MO) treatment at 26 ℃ for 24 hours and concentrated by polyethylene glycol (PEG, MW 6,000, Sigma-Aldrich) precipitation.
Reviewer comment 4: Figure 3: the authors set up a few experiments with different moi to examine the sensitivity of BHK-S to FMDV. However, the authors failed to add the conventional adhesive culture of BHK-21 cells at the same cellular density as a control. The fact that yield of FMDV is reduced over time may suggest that suspension and serum-free adaptation reduced the expression of some molecules (i.e., integrins) required for virus entry.
Author response 4: We agree with your comment. It would be better if we could compare the conventional adhesive BHK-21 cells to BHK-S at the same cellular density. Instead, we compared FMDV productivity between those cells at the same culture volume in Fig. 1D. Additionally, the reduced yield of FMDV as the harvest time is delayed may be due to proteolysis by several kinds of enzymes from the lysed cells. Although we did not include our preliminary data that was collected up to 12 hours post-infection, the maximal productivity was observed at 16 hours post-infection when we used the FMDV/O/JC/2014 strain.
Reviewer comment 5: In all graphs the authors need to indicate how many experiments were conducted and whether the error bars one standard deviation of uncertainty or one standard error.
Author response 5: The experiments were conducted by triplicates (Fig. 1D, Fig 2, Fig 3, Fig 4, and Fig 5). The error bars represent one standard deviation, which was mentioned in every figure legend.
e.g. Figure 1. Morphologies of adherent BHK-21 (C13) cells in a T-flask (A) or BHK-S cells as suspension cultures in a 125 mL Erlenmeyer flask (B). The stability of the BHK-S cells was monitored for 3 weeks (C). The cell viability exceeded 95% at each passage. Production of FMDV by BHK-21 and BHK-S cells was compared using the same culture volume (D). Each experiment was carried out in triplicate. Data expressed as mean ± standard deviation.
Reviewer 2 Report
The manuscript “Production of a foot-and-mouth disease vaccine antigen using suspension-adapted BHK-21 cells in a bioreactor” describes the adaptation of BHK21 cell line to grow in suspension in serum-free medium and their evaluation for foot-and-mouth disease virus (FMDV) infection and inactivated vaccine production.
The topic of the manuscript is of high interest, the use of vaccination to contrast FMDV epidemics is very important, and to achieve eradication, especially in those countries where FMDV is still endemic, a cost-effective high-quality vaccine is essential.
The manuscript has been well organized and the study has been well carried out and conceived. The results are consistent with the discussion and the conclusion.
I have only one concern: the authors proved and compared the production yield of 146S FMDV antigen however they do not evaluate if its antigenic properties are maintained. As the authors well explained in the discussion the changes in the adhesive properties can influence FMDV infectivity, FMDV has a high mutation rate thus some kind of amino acid mutation can occur in the capsid proteins leading to a highly infective virus but antigenically different from the original virus.
Minor remarks:
The 146S terminology is used by the authors for the first time in Materials and Methods, I think that it would be better to cite it and its significance in the introduction
Line 113: Please could the authors describe better the method of the cell-adhesion assay, especially its rational
Line 151: “146S” there is a space and a dot between 146 and S
Line 152: please clarify how the virus has been inactivated
Line 282-283: I do not understand “without 20% FM-1” may be a mistake?
Author Response
Reviewer : The manuscript “Production of a foot-and-mouth disease vaccine antigen using suspension-adapted BHK-21 cells in a bioreactor” describes the adaptation of BHK21 cell line to grow in suspension in serum-free medium and their evaluation for foot-and-mouth disease virus (FMDV) infection and inactivated vaccine production.
The topic of the manuscript is of high interest, the use of vaccination to contrast FMDV epidemics is very important, and to achieve eradication, especially in those countries where FMDV is still endemic, a cost-effective high-quality vaccine is essential.
The manuscript has been well organized and the study has been well carried out and conceived. The results are consistent with the discussion and the conclusion.
Reviewer comment 1: I have only one concern: the authors proved and compared the production yield of 146S FMDV antigen however they do not evaluate if its antigenic properties are maintained. As the authors well explained in the discussion the changes in the adhesive properties can influence FMDV infectivity, FMDV has a high mutation rate thus some kind of amino acid mutation can occur in the capsid proteins leading to a highly infective virus but antigenically different from the original virus
Author response 1: We totally agree with your opinion. Although we did not conduct animal experiments to prove its antigenic properties with these antigens, we have already confirmed that the antigenicity of the FMDV 146S antigens produced by BHK-S in serum-free media did not show significant changes to that of the original viruses through our previous studies with other strains as follows.
- Microbiol. (2020) 248: 108802
- Vaccine (2020) 38: 1120-1128
- Microbiol. (2019) 229: 124-129
Reviewer comment 2: The 146S terminology is used by the authors for the first time in Materials and Methods, I think that it would be better to cite it and its significance in the introduction
Author response 2: We added the information about the significance of 146S at lines 86-88 on page 2 per your comment.
‘FMDV can be divided into several particles (146S, 75S and 12S) and 146S is intact virions [17]. Since 146S antigens are the most efficient immunogens in the FMD vaccine, 146S anti-gens were also quantified in each production scale [18].’
Reviewer comment 3: Line 113: Please could the authors describe better the method of the cell-adhesion assay, especially its rational.
Author response 3: The cell adhesion assay has been clarified in the methods section at lines 119–135.
2.3. Cell-adhesion assay
Cell adhesion assay was performed under sterile conditions using the CytoSelect 48-Well Cell Adhesion Assay, ECM Array (CBA-070, Cell Biolabs Inc., CA, USA). Briefly, the plates coated with various ECM components (fibronectin, collagen I, collagen IV, fi-brinogen, and laminin or bovine serum albumin as a negative control) were warmed up at room temperature for 10 min. 1 × 106 cells/mL cells were then prepared in serum-free me-dium, 150 μl of cell suspension was added to appropriate wells, and the plates were in-cubated for 60 min at 37°C in a humidified atmosphere in the presence of 5% CO2. The medium was removed, and each well was washed 4 times with 250 μl phos-phate-buffered saline (PBS). The PBS was removed and 200 μl of the cell-staining solution (provided in the kit) was added to each well and the plates were incubated for 10 min at room temperature. After incubation, the cell-staining solution was removed, and each well was washed 4 times with 500 μl deionized water. The final wash was removed, and the wells were dried. Then, 200 μl extraction solution (provided in the kit) was added to each well, and the plates were incubated for 10 min on an orbital shaker at room temperature. Finally, 150 μl of each sample was transferred to each well in a 96-well microtiter plate, and the optical density at 560 nm (OD560 nm) was measured using a Synergy HT instru-ment (BioTek, USA). Each experiment was carried out in triplicate.
Reviewer comment 4: Line 151: “146S” there is a space and a dot between 146 and S
Author response 4: In previous papers, such as the ones that we referred to, there is no space or dot between 146 and S when indicating the sedimentation value.
- Vaccines (2021) 9, 387
Selection of Vaccine Candidate for Foot-and-Mouth Disease Virus Serotype O Using a Blocking Enzyme-Linked Immunosorbent Assay
- PLOS ONE (2016) March 1 https://doi.org/10.1371/journal.pone.0149569
Quantitative Detection of the Foot-And-Mouth Disease Virus Serotype O 146S Antigen for Vaccine Production Using a Double-Antibody Sandwich ELISA and Nonlinear Standard Curves
- Vaccines (2020) 8, 583
Targeted Modification of the Foot-And-Mouth Disease Virus Genome for Quick Cell Culture Adaptation
Reviewer comment 5: Line 152: please clarify how the virus has been inactivated
Author response 5: We added information about the inactivation method at line 165-166 on page 4 per your comment.
‘The clarified virus culture supernatant of 50 mL was inactivated with 3 mM binary-ethylenimine (Sigma-Aldrich, St. Louis, MO) treatment at 26 ℃ for 24 hours and con-centrated by polyethylene glycol (PEG, MW 6,000, Sigma-Aldrich) precipitation.’
Reviewer comment 6: Line 282-283: I do not understand “without 20% FM-1” may be a mistake?
Author response 6: We apologize for the confusion. The maximum viable cell concentration was increased when the amount of FM-1 (v/v) was increased, but the MVCC was dramatically reduced in media containing 20% (v/v) FM-1. This has been clarified at lines 318–320 on page 9.
‘As the concentration of FM-1 increased from 5% to 20%, the MVCC were increased dose dependent manner, whereas 20% FM-1 dramatically suppressed cell growth.’